# Predicting evolution from the shape of genealogical trees

Richard A Neher[1]*, Colin A Russell[2], Boris I Shraiman[3]*

[1]Evolutionary Dynamics and Biophysics, Max Planck Institute for Developmental Biology, Tübingen, Germany; [2]Department of Veterinary Medicine, University of Cambridge, Cambridge, United Kingdom; [3]Kavli Institute for Theoretical Physics, University of California, Santa Barbara, Santa Barbara, United States

**Abstract** Given a sample of genome sequences from an asexual population, can one predict its evolutionary future? Here we demonstrate that the branching patterns of reconstructed genealogical trees contains information about the relative fitness of the sampled sequences and that this information can be used to predict successful strains. Our approach is based on the assumption that evolution proceeds by accumulation of small effect mutations, does not require species specific input and can be applied to any asexual population under persistent selection pressure. We demonstrate its performance using historical data on seasonal influenza A/H3N2 virus. We predict the progenitor lineage of the upcoming influenza season with near optimal performance in 30% of cases and make informative predictions in 16 out of 19 years. Beyond providing a tool for prediction, our ability to make informative predictions implies persistent fitness variation among circulating influenza A/H3N2 viruses.

## Introduction

A general method to predict the evolutionary trajectories of asexual populations would be extremely valuable for understanding the population dynamics of pathogens or of malignant cells. For example, the vaccine against seasonal influenza needs to be updated frequently since virus populations evolve to evade increasing immunity among humans (*Hampson, 2002*; *Nelson and Holmes, 2007*). Reliable prediction of the strains most likely to circulate in the upcoming season, and particularly the ability to predict antigenic change, would be transformative to the vaccine strain selection process.

Predictability from genetic sequence data requires heritable fitness variation among the sampled sequences. Neutral evolution - population dynamics in the absence of selective pressure - is by definition unpredictable: all sequences are equally fit. Yet even when selection determines the success of individual lineages, predictability depends on the effect size of fitness-altering mutations. Two competing scenarios of adaptive evolution are illustrated in *Figure 1*. If evolution proceeds via rare mutations with large phenotypic effects, the population is homogeneous in fitness most of the time (*Figure 1A*). In this case large effect mutations can convert any genome into the fittest in a single generation. Prediction from sequence alone is only possible if the time of sampling happens to be during a brief sweep of a large effect mutation. In contrast, continuous accumulation of small effect mutations (*Figure 1B*) results in a gradual change in fitness of lineages and persistent variation in fitness (*Tsimring et al., 1996*). A genealogical tree then potentially contains predictable patterns: the fitness of most lineages decreases over time (movement to the left in *Figure 1*), due to a changing environment or the accumulation of weakly deleterious mutations. Only a few adapt rapidly enough to stay among the most fit in the population (*Rouzine et al., 2003*; *Brunet et al., 2007*; *Desai and Fisher, 2007*; *Hallatschek, 2011*; *Goyal et al., 2012*; *Desai et al., 2013*; *Neher and Hallatschek, 2013*) and thus have a chance to continue into the future.

*For correspondence: richard. neher@tuebingen.mpg.de (RAN); shraiman@kitp.ucsb.edu (BIS)

**eLife digest** When viruses multiply, they copy their genetic material to make clones of themselves. However, the genetic material in the clone is often slightly different from the genetic material in the original virus. These mutations can be caused by mistakes made during copying or by radiation or chemicals. Further mutations arise when the clones multiply, which means that, after many generations, there will be quite large differences in the genetic material carried by many members of the population. Most mutations have little or no effect on the 'fitness' of an individual - that is, on its ability to survive and multiply - but some mutations do have an influence.

Some viruses, like seasonal influenza (flu) viruses, can mutate so rapidly that the most common strains change from year to year. This is why new flu vaccines are needed every year. To date most attempts to predict the evolution of seasonal flu viruses have focused on identifying specific features within the genetic sequences that might indicate fitness. However, such approaches require lots of information about the viruses, and this information is often not available.

To address this problem, Neher, Russell and Shraiman have developed a more general method to predict fitness from virus genetic sequences. First, a 'family tree' for a virus population - which shows how each strain of the virus is related to other strains - was constructed by comparing the genetic sequences.

The next step was based on the observation that as long as differences in fitness arise from the accumulation of multiple mutations, the branching structure of this family tree will bear a visible imprint of the natural selection process as it unfolds. Using this insight and methods borrowed from statistical physics, Neher et al. then analyzed the shape and branching pattern of the tree to work out the fitness of the different strains relative to each other.

Neher et al. tested the method using historical influenza A virus data. In 16 of the 19 years studied, the family tree approach made meaningful predictions about which viruses were most likely to give rise to future epidemics. The ability to predict influenza virus evolution from tree shape alone suggests that influenza virus evolution may be more predictable than previously expected.

In the specific context of human seasonal influenza A/H3N2 viruses, the study of their antigenic evolution has identified specific amino-acid substitutions with large phenotypic effects (*Koel et al., 2013*), that have been responsible for the observed stepwise replacement of antigenic variants over time (*Smith et al., 2004*). Yet, the evolution of seasonal influenza viruses is also marked by the continuous accumulation of mutations that have small or no antigenic effects but nevertheless potentially affect fitness (*Bhatt et al., 2011*; *Strelkowa and Lässig, 2012*), for example compensatory or permissive mutations (*Gong et al., 2013*). Previous attempts at predicting the evolution of seasonal influenza viruses have tried to identify molecular signatures that are predictive of future success (*Bush et al., 1999*) or used clustering approaches based on amino acid sequences (*Plotkin et al., 2002*). Recently, *Łuksza and Lässig (2014)* constructed an explicit fitness model based on sequence data from the hemagglutinin (HA1) surface protein. The utility of these explicit models depend on the availability of extensive historical data or a detailed understanding of the influenza virus sequence-to-fitness map.

Rather than constructing an explicit fitness model, which is currently impossible for most organisms, we developed a general algorithm to infer fitness from the shape of reconstructed genealogical trees without using any molecular information. Our approach is based on a simple idea: since high (Malthusian) fitness implies many offspring, which in turn implies branching, the shape of the tree can be exploited to infer fitness (*Dayarian and Shraiman, 2014*). Here, we developed a quantitative model of fitness dynamics on genealogical trees, which is based on recent progress in understanding the statistical structure of genealogies in adapting populations (*Neher and Hallatschek, 2013*). Following *Neher and Hallatschek (2013)*, our model assumes: 1) that the population is under persistent directional selection and 2) fitness changes along lineages in small steps through the continuous accumulation of small effect mutations (*Figure 1B*). This fitness model resembles the well-known infinitesimal model of quantitative genetics (*Falconer and Mackay, 1996*) in the sense that many small effect mutations give rise to a bell-shaped fitness distribution on which selection acts (*Neher, 2013*). However, the infinitesimal model itself provides no insight into the relationship between the structure of genealogical trees and fitness: this insight stems from the more recent work on the dynamics of adaptation in

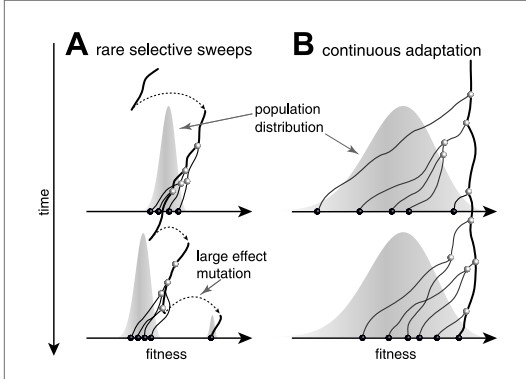

**Figure 1**. Genealogies in adapting populations. (**A** and **B**) illustrate the genealogy of two successive samples embedded into the (Malthusian) fitness distribution of the population indicated in grey. In absence of adaptive mutations, fitness declines due to a changing environment or accumulation of deleterious mutations. Only one lineage (thick line) persists from first sample to second sample. (**A**) Evolution proceeds via rare large effect mutations (dashed arrows) that occur in a population with little fitness variance. All individuals are roughly equally likely to pick up the large effect mutation, rendering evolution unpredictable from sequence data alone. (**B**) Conversely, if adaptation is due to many small effect mutations, the successful lineage (thick) is always among the most fit individuals. Being able to predict relative fitness therefore enables to pick a progenitor of the future population.

large asexual populations (*Tsimring et al., 1996*; *Rouzine et al., 2003*; *Desai and Fisher, 2007*; *Desai et al., 2013* ; *Neher and Hallatschek, 2013*) and in populations with occasional reassortment (*Neher and Shraiman, 2011*). After testing the algorithm on simulated data we apply our algorithm to historical data on human seasonal influenza A/H3N2 virus hemagglutinin sequences. Despite multiple confounding factors – discussed below – we find that our algorithm makes informative predictions about influenza virus evolution.

## Results

### The fitness distribution on a tree

Intuitively, we expect that an exceptionally fit internal node in a genealogical tree will be at the root of a rapidly branching, and hence expanding, clade (e.g. node 2 in *Figure 2A*). Similarly, extant individuals with high fitness are likely to be recent descendants of internal nodes with high fitness (e.g. node 3 in *Figure 2A*). By tracing fitness along lineages and integrating across the tree, the algorithm described below makes this intuition precise and quantitative.

As input, our algorithm requires a genealogical tree, e.g. a tree reconstructed from a sample of genomic sequences. For a given tree $T$, we derived the joint probability distribution $P(\mathbf{x}|T)$ for the fitnesses $\mathbf{x} = x_0, x_1, \ldots$ of all internal nodes (corresponding to reconstructed ancestral sequences) and external nodes (corresponding to the sampled genomes). Fitness $x_i$ of each node $i$ is measured relative to the population mean fitness at the time when the corresponding individual was sampled. $P(\mathbf{x}|T)$ is given by a product of *propagators* $g(\cdot|\cdot)$ for each branch

$$P(\mathbf{x}|T) = \frac{p_0(x_0)}{Z(T)} \prod_{i=0}^{n_{\text{int}}} g(x_{i_1}, t_{i_1}|x_i, t_i) g(x_{i_2}, t_{i_2}|x_i, t_i), \qquad (1)$$

where $p_0(x)$ is the fitness distribution in the population (see 'Materials and methods' for details) and the index $i$ runs from 0 (the root) through all $n_{\text{int}}$ internal nodes. The indices $i_1$ and $i_2$ denote the two children of node $i$, while $Z(T)$ ensures normalization of the distribution. *Eq. (1)* has a structure similar to the expression for the likelihood of sampled sequences, given a tree $T$, defined in phylogenetic analysis (*Felsenstein, 2003*). The main difference is that instead of defining the probability of mutation from one character state to another, the branch propagator $g(x_j, t_j|x_i, t_i)$ describes the likelihood of the lineage to connect an ancestor with fitness $x_i$ at time $t_i$ to a child with fitness $x_j$ at a later time $t_j$ (child in sense of a subclade in the tree, rather than direct offspring). Note that a branch connecting nodes $i$ and $j$ implies that all sampled descendants of $i$ are also descendants of $j$, i.e., the 'branch does not branch'. This non-branching condition is part of the branch propagator which therefore depends on the fraction $\omega$ of the total population that is represented in the sample (see 'Materials and methods' for details).

*Figure 2A* illustrates the propagator as function of child fitness $x_j$, which describes the fitness distribution of children, conditioned on ancestral fitness $x_i$. At small $\Delta t = t_j - t_i$, the distribution is peaked around the ancestor. At long times, memory of ancestral fitness is lost and the propagator approaches the population distribution. Backwards in time, $g(x_j, t_j|x_i, t_i)$ describes (using the Bayesian inversion formula [*Felsenstein, 2003*]) the fitness distribution of the ancestor $i$ given a sampled child with fitness $x_j$ at time $t_j$. Far in the past, the ancestor fitness distribution converges to a narrow peak in the high

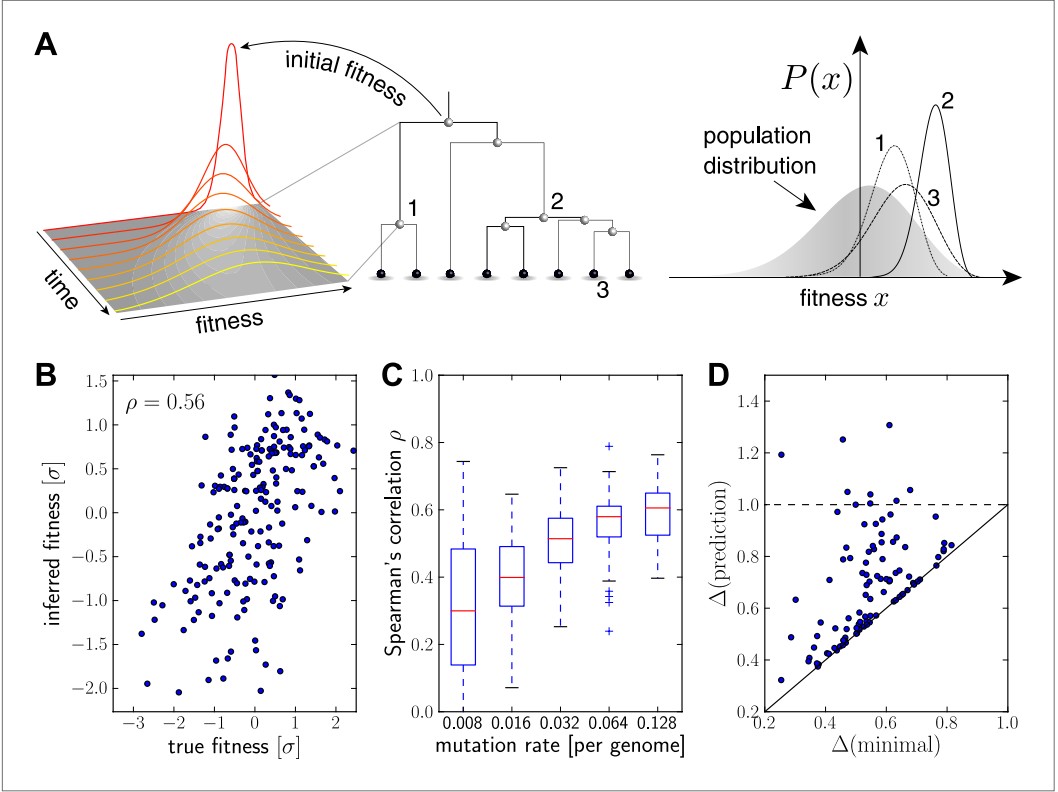

**Figure 2**. Inferring fitness from genealogical trees. (**A**) The inference algorithm is based on branch propagators associated with each branch of the reconstructed tree (middle). Branch propagators characterize the fitness distribution of child nodes given the fitness of the ancestral node (left). The internal node 2 would have higher marginal fitness estimate (right) than node 1, as node 2 has more children. The inferred distribution of the fitness of the external node 3 has broadened along the branch from node 2. (**B–D**) Analysis of simulated data. Panel B shows for a typical example that inferred fitness is well correlated with the true fitness with a rank correlation coefficient $\rho = 0.56$. This correlation increases with increasing mutation rate as shown in panel C for 100 simulated data sets each (boxes cover the interquartile range, red lines indicate the median). Panel D shows that the sequence with the highest inferred fitness tends to be similar to the population 200 generations in the future. Both axis show the average Hamming distance to the future population between the predicted and the post-hoc optimal sequence on the $y$ and $x$-axis, respectively, for 100 simulated data sets. Both distances are relative to the average distance between the present and future population. Parameters: $N = 20000$, $n_A = 0.08$, $\Gamma = 0.2$, $u = 0.064$ (B,D).
The following figure supplements are available for figure 2:

**Figure supplement 1**. Predictability increases with genetic diversity.

**Figure supplement 2**. Prediction from continuously sampled sequences.

fitness tail (*Rouzine and Coffin, 2007*; *Neher and Hallatschek, 2013*). See 'Materials and methods' for a more detailed discussion.

The fitness dynamics along a lineage resemble a random walk on which each step corresponds to a mutation with a certain effect on fitness. This walk is biased towards high fitness by selection, which makes fitter lineages more likely to survive and eventually be sampled. If many mutations contribute, the dynamics of fitness along branches can be approximated by selection-biased diffusion (SBD) as described in 'Materials and methods', *Equation (9)* – *Equation (11)*. The fitness diffusion constant of a branch is given by $D = u\langle s^2 \rangle / 2$, where $u$ is the genome wide mutation rate, and $\langle \cdot \rangle$ denotes the average over the effect sizes of mutations (*Tsimring et al., 1996*). Fitness diffusion and stochasticity due to finite populations determine the fitness variance $\sigma^2$ in the population (*Cohen et al., 2005*).

Based on the SBD approximation derived in 'Materials and methods', we implemented a program that numerically solves for the branch propagator and, by going up and down the tree using a 'Message

Passing' (similar to dynamic programming) technique (*Mézard and Montanari, 2009*), calculates the marginal fitness distribution for each node as illustrated in *Figure 2A*, for details see 'Materials and methods'.

## Fitness inference is insensitive to model assumptions

To explore the extent to which the idealized SBD model assuming infinitesimal mutations is able to infer fitness when evolution happens via discrete mutations, we simulated a simple model of evolution with fixed fitness variance ($\sigma = 0.03$) (*Zanini and Neher, 2012*). In order to mimic adaptive evolution in a changing environment we introduced sites in the simulated genome that allow for beneficial mutations at rate $n_A = 0.02, \ldots, 0.16$ per generation in a genome otherwise dominated by deleterious mutations. Every 200 generations, we took a random sample of sequences from the simulated population. We recorded the fitness of each sampled sequence, which we will compare with our inferences below.

In order to apply the fitness inference method to a reconstructed tree, we needed to parameterize the model and convert branch length measured as similarity between sequences into time. When measuring time in units of $\sigma^{-1}$, the SBD model has only one free dimensionless parameter $\Gamma = D\sigma^{-3}$ that describes the relative importance of selection and stochastic processes. $\Gamma$ is inversely proportional to the square root of the logarithm of the population size and hence does not vary greatly (*Tsimring et al., 1996*; *Cohen et al., 2005*). We used $\Gamma = 0.2$ and $0.5$ corresponding to moderate and more rapid diffusion relative to selection, respectively. Coalescent theory of adapting population connects pairwise sequence similarity to $\Gamma$. The choice of $\Gamma$ fixes the conversion from branch length to time via *Equation (20)* (*Neher and Hallatschek, 2013*). In addition to $\Gamma$ we need to fix $\omega$. Since we used a sample of 200 sequences out of a total of $N = 20000$ sequences, $\omega = 0.01$ (ultimately, $\omega/\sigma$ enters the algorithm, see 'Materials and methods'). Using these parameters, we applied our method to a reconstructed tree and report the mean posterior fitness as 'inferred fitness' for each internal and external node.

*Figure 2B* shows the inferred vs true fitness for a typical simulation. The rank order of fitness is well predicted (Spearman's correlation coefficients around $0.5$). *Figure 2C* shows that fitness rankings improve with increasing mutation rates. This is expected, since increased mutation rates correspond to a larger number of mutations that contribute to fitness and make the SBD model a better approximation. This behavior is consistent across different rates of adaptive mutations and depends weakly on our choice of $\Gamma$ (*Figure 2—figure supplement 1*). Large $\Gamma$ performs better at low mutation rates when fitness diversity is dominated by only a few mutations, corresponding to more rapid fitness diffusion relative to selection and coalescence.

## High inferred fitness predicts progenitor sequences

Next, we asked whether sequences that we predict to have high fitness are close in sequence to the progenitor lineage of future populations. *Figure 2D* shows the Hamming distance $\Delta(\mathrm{prediction})$ of the sequence of the individual with the highest fitness estimate to the population 200 generations in the future vs the $\Delta(\mathrm{minimal})$ for the post-hoc optimal pick. The measure $\Delta(\mathrm{sequence})$ is normalized to the average Hamming distance between the present and future population. In 40 out of 100 simulations, the top-ranked sequence is an almost optimal pick (points close to the diagonal in *Figure 2D*). In 8 out of 100 cases, the prediction is better than a random pick (points below the dashed line *Figure 2D*).

The fitness inferences shown in *Figure 2B–C* used 200 sequences sampled from the same generation. However, the influenza data to which we apply our algorithm below is continuously sampled throughout the year. In *Figure 2—figure supplement 2* we reproduce panels B–C using 200 sequences sampled from the simulation over a time interval of 100 generation. This gives highly similar results.

### Local branching density as a heuristic ranking

In general, faithful inference of the posterior fitness distribution requires numerical solution for the branch propagators and knowledge of the parameters $\Gamma$ and $\omega/\sigma$. We observed, however, that the ranking of nodes by fitness and the prediction of progenitor lineages depends little on these parameters. This insensitivity suggests that the fitness ranking depends primarily on a more universal quantity on which the inference algorithm builds.

In 'Materials and methods', we show that the fitness estimates of internal nodes increase with the total branch length downstream of these nodes–at least for short time periods. The downstream tree length acts as a "polarizer" that pushes the fitness distribution of the node away from the population mean towards high fitness. For given number of descendants, the length of a subtree is maximal if it is

star-like. This is intutive, as star-like subtrees indicate rapid branching (or multiple mergers backwards in time) which is expected for high fitness nodes. Conversely, prolonged absence of branching of a lineage indicates relatively low fitness.

If fitness changes gradually along lineages, high fitness of a node will coincide with both upstream and downstream branching–at least within a certain neighborhood of the tree. The relevant size of the neigborhood will depend on how rapidly fitness decorrelates along lineages. Based on this intuition, we developed a model-independent heuristic ranking algorithm: for each internal and terminal node $i$, we calculate a *local branching index (LBI)* $\lambda_i(\tau)$ defined as total surrounding tree length exponentially discounted with increasing distance from the focal node. The scale $\tau$ of the exponential discounting corresponds to the size of the relevant tree neighborhood or the time over which fitness is 'remembered' across the tree. Within the SBD model, $\tau$ corresponds to the equilibration time scale of lineage fitness in the high fitness tail, which is of the order $T_c/\sqrt{\log N}$, where $T_c$ is the coalescence time scale (*Neher and Hallatschek, 2013*).

The LBI can be efficiently calculated with the same message passing techniques we used to calculate the posterior fitness distribution. Remarkably, rankings obtained by this simple heuristic are almost as accurate as fitness inference using the more complex SBD model. *Figure 3* shows Spearman's correlation coefficient of $\lambda_i(\tau)$ with true fitness as a function of pairwise difference for different memory time scales $\tau$ and compares it to the ranking via mean inferred fitness. The heuristic $\lambda_i(\tau)$ not only correlates well with true fitness in simulations but sequences with the highest $\lambda_i(\tau)$ also tend to be close to the progenitor of future populations (*Figure 3—figure supplement 1*). Comparing the performance of the LBI to the full fitness inference in *Figure 3*, we concluded that a neighborhood size should be $\tau \approx 0.0625$ of the average pairwise distance in the sample.

## Prediction of seasonal influenza A/H3N2 progenitor lineages

Having validated our algorithm on simulated data and presented a model independent method to rank sequences, we attempted to predict progenitor sequences of seasonal influenza A/H3N2 viruses. We used samples of influenza A/H3N2 virus hemagglutinin (HA1) sequences from one year (May–February, Asia and North America, at most 100 sequence from each region) to predict the closest relative of the population circulating in the following (northern hemisphere) winter (October–March, Asia and North America) for the years 1995–2013. All HA1 domain sequences used for our analysis came from the public domain and are available from Influenza Research Database (www.fludb.org (*Squires et al., 2012*)). Next, we built maximum likelihood trees using fasttree (*Price et al., 2009*), collapsed zero-length branches into polytomies, and ranked external and internal nodes using the LBI. We set the memory time scale to $\tau = 0.0625$ in units of average pairwise distance as suggested by the simulation data. Details of the data sets used for making predictions and discussion of potential biases are given in 'Materials and methods'. *Figure 4A&B* show example trees of the prediction and test sets for 2007.

*Figure 4C* shows the nucleotide distance of our prediction to the A/H3N2 virus population of the next season, both for the top-ranked internal and external node of each year. Using the highest ranked external node (*Figure 3C*, black squares) is similar to using the highest ranked internal node (*Figure 3C*, red diamonds) in all years but 1997. The highest ranked internal node predict years 1997–1999, 2003, 2006–2009, and 2013, reasonably well. Notably, they fail in 1995, 1996, and 2002, while being of intermediate accuracy in the remaining years. The dependence of the prediction accuracy on the neighborhood size $\tau$ is shown in *Figure 4—figure supplement 1*. We also predicted successful progenitor strains using the fitness inference based on the SBD model which yields results very similar to the ranking by LBI–sometimes slightly better, sometimes worse depending on parameter choice.

We compared our predictions to vaccine strain predictions obtained by *Łuksza and Lässig (2014)* who predict progenitors of future epidemics as we do here, albeit using an influenza specific model with four parameters, two of which are trained for each individual prediction on data from several preceding years. On average, using the same time cutoffs for prediction (February to predict October) as we used above, Łuksa and Lässig achieve an accuracy comparable to our parameter-free ranking based (see *Figure 4—figure supplement 2*). Interestingly, these two rather different approaches yield very similar predictions on a year to year basis. One potential explanation for this concordance is an ad hoc aspect of Łuksa and Lässig's model meant to capture epistatic interactions: the total number of synonymous mutations downstream of each clade is used as an additional predictor. The number of synonymous mutations is strongly correlated with tree length and hence with $\lambda_i(\tau)$.

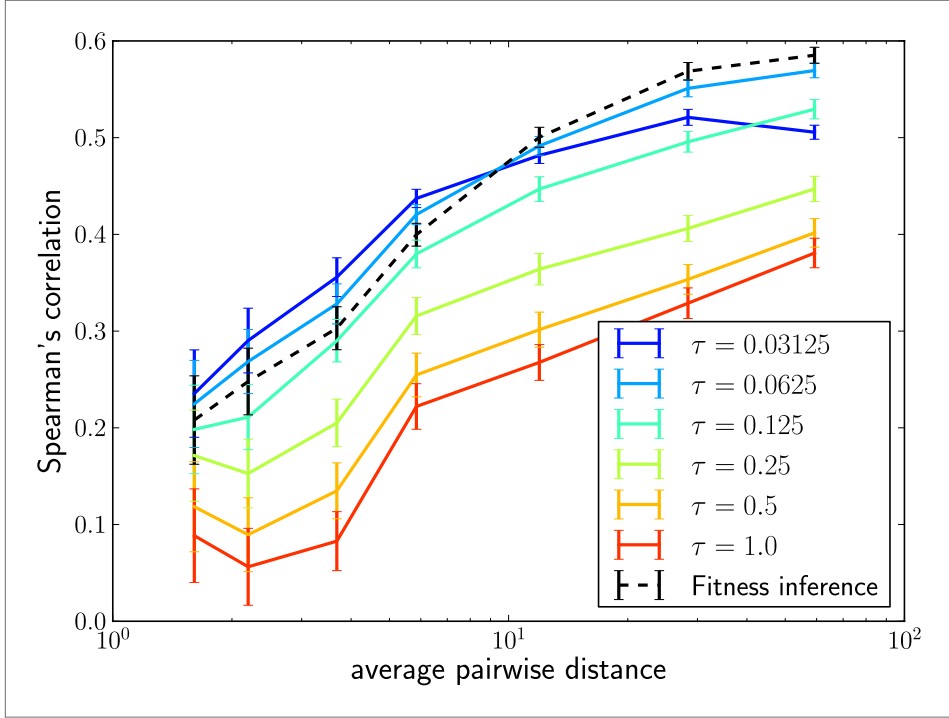

**Figure 3**. Local tree length as a fitness ranking. Rank correlation between the true fitness and the LBI $\lambda_i(\tau)$ is shown as a function of pairwise diversity in the sample. Different curves correspond to different neighborhood sizes $\tau$, which is measured in units of the average pairwise distance.

The following figure supplement is available for figure 3:

**Figure supplement 1**. The LBI predicts progenitor sequences.

To quantify prediction quality across years, we define the distance measure $d = (\Delta(\text{prediction}) - \Delta(\text{minimal}))/(1 - \Delta(\text{minimal}))$ such that an optimal prediction has $d = 0$ and a random pick has $d = 1$. The average of $d$ over all years is denoted by $\overline{d}$. *Figure 5* shows bootstrap distributions of $\overline{d}$ for our methods and compares it to *Łuksza and Lässig (2014)* as well as two naive prediction methods: (i) a growth rate estimate of individual clades obtained by fitting an exponential curve to the fraction of the total sequences that are part of this clade in three time intervals between May and February, and (ii) the sequence of the most advanced node in a ladderized tree. Predictions with the method described here and by *Łuksza and Lässig (2014)* are comparable within errorbars, while the two naive estimators do substantially worse on average. The dependence of the average predictive power of the LBI on the neighborhood size $\tau$ is shown in *Figure 5—figure supplement 1*.

### Inferred fitness increases are associated with epitope mutations

Changes in fitness along branches can be associated with the types of mutations on those branches. We found that branches corresponding to the top quartile of differentials of $\lambda_i(\tau)$ are enriched for non-synonymous substitutions over synonymous mutations. Restricting non-synonymous mutations to the epitopes A–D (used in (*Łuksza and Lässig, 2014*) and defined in (*Shih et al., 2007*)) increases this enrichment to approximately 2-fold, see *Table 1*. Further restriction to the 7 loci identified Koel et al. increases the enrichment slightly, but their number is small and the power to detect additional enrichment is low. These findings are consistent with the notion that influenza evolution is driven by antigenic novelty (*Wiley et al., 1981*; *Hampson, 2002*; *Smith et al., 2004*) and provide independent confirmation of the power of the sequences ranking and fitness inference algorithm.

### Discussion

Starting with a model of adaptive evolution, we developed a probabilistic description of the fitness dynamics on genealogical trees and presented an algorithm to infer fitness of individual nodes in the

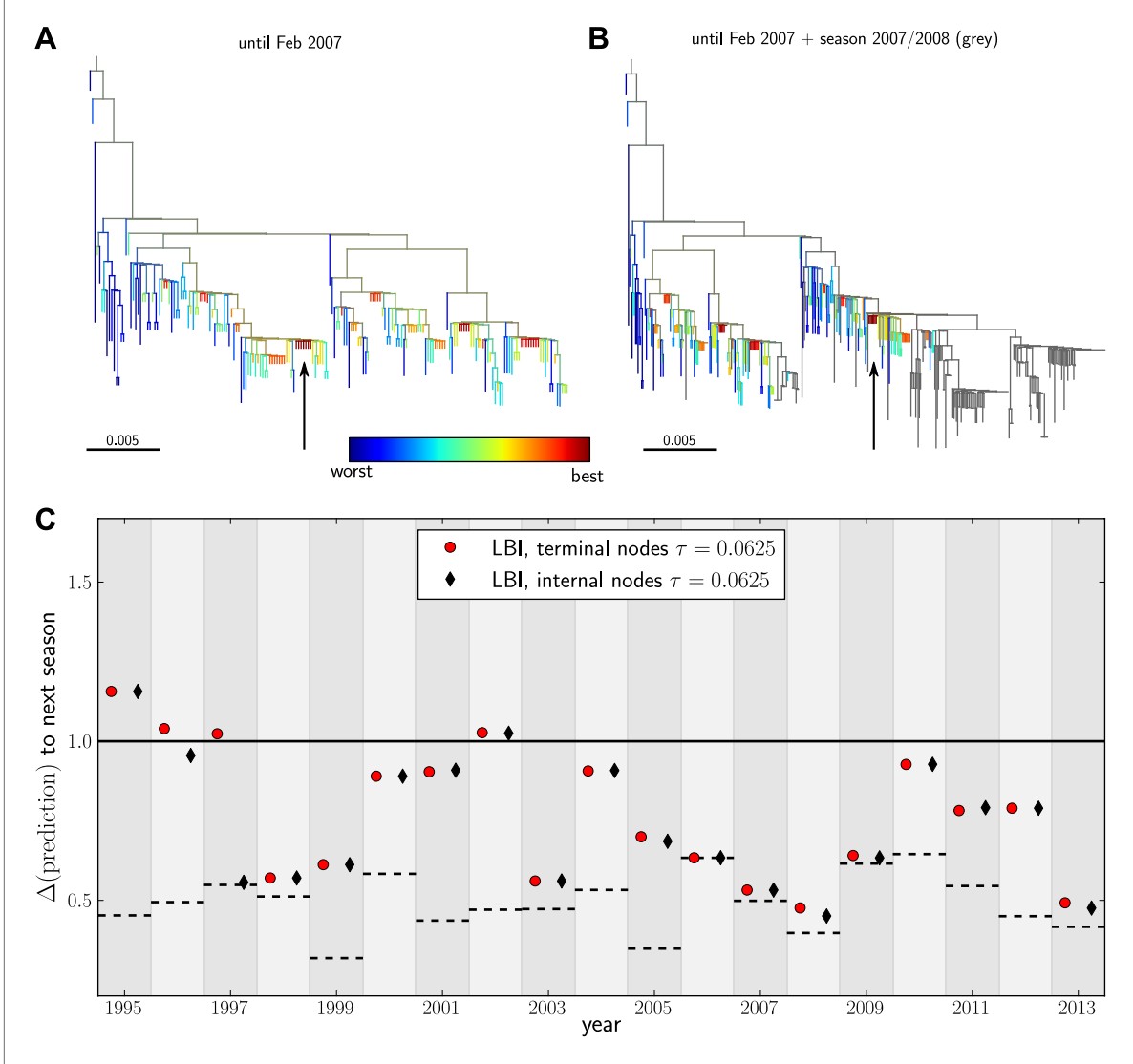

**Figure 4**. Predicting the evolution of seasonal influenza A/H3N2 viruses. (**A**) A genealogical tree of a sample of HA1 sequences from May 2006 to end of February 2007. Nodes are colored according to our fitness ranking $\lambda_i(\tau)$. The highest ranked node is marked by a black arrow. (**B**) A tree of the same sequences from (**A**) (colored) and sequences from October 2007 to end of March 2008 (in grey). Our algorithm successfully predicts a sequence genetically close and directly ancestral to viruses circulating the following winter. (**C**) For each year from 1995 to 2013 we predicted a progenitor sequence and calculated its nucleotide distance to the A/H3N2 population of the following winter. Predictions based on terminal or internal sequences are very similar. The figure shows the average $\Delta(\mathrm{prediction})$ of 50 runs using subsamples of the data. A random pick from the prediction set corresponds to the solid line at 1. The dashed lines indicate the optimal extant sequence at time of prediction. The distance of the dashed line from the line at 1 indicates the closeness of the optimal extant sequence to future populations.

The following figure supplements are available for figure 4:

**Figure supplement 1**. Variation of predictions upon variation of the memory time scale of the LBI $\lambda_i(\tau)$.

**Figure supplement 2**. Comparison to predictions by *Łuksza and Lässig (2014)*.

**Figure supplement 3**. High LBI predicts clade expansion.

tree. We validated this algorithm using trees reconstructed from simulated sequences and showed that the sequence with the highest inferred fitness tends to be a close match to the progenitor of future populations. Analysis of the model revealed that a simple quantity–the local branching index (LBI)–determines the fitness estimates and can be used to rank sequences by fitness with similar

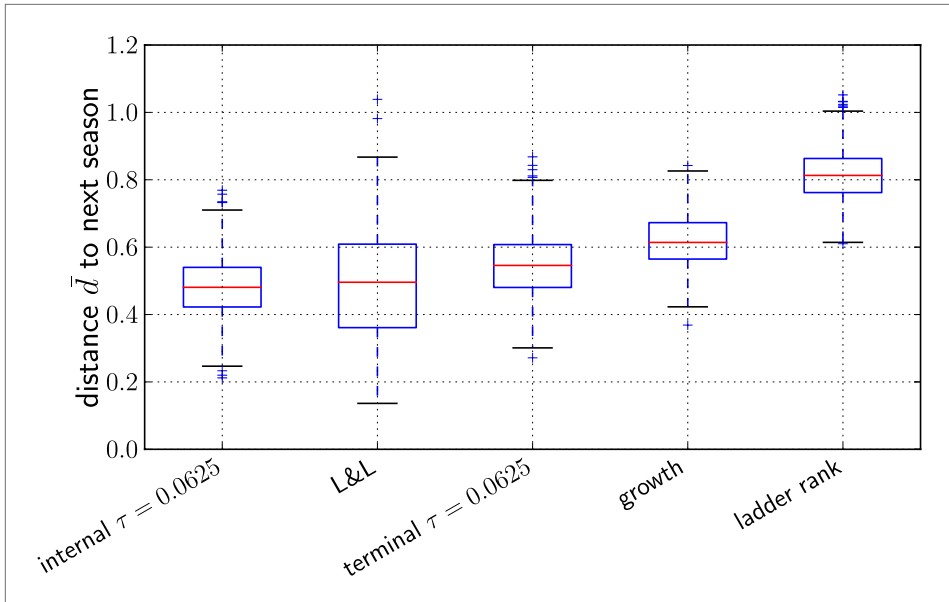

**Figure 5**. Comparison of predictors. Transformed genetic distance $\bar{d}$ averaged over 1000 bootstrap samples (bootstrapping years) to the next influenza season. We compared our method using the sequence of the top ranked internal node, external node, the predictions by *Łuksza and Lässig (2014)*, the ancestral sequence of clades with the largest estimated growth rate, and the sequence of the most 'advanced' node in a ladderized tree.

The following figure supplement is available for figure 5:

**Figure supplement 1**. Dependence of prediction accuracy on $\tau$.

accuracy as the full fitness inference algorithm. The only parameter of the LBI is the size of the neighborhood on the tree and a suitable value can be chosen from simulated data.

Our fitness inference framework is based on the selection-biased diffusion model that assumes evolution proceeds via accumulation of many small effect mutations. As expected, its predictive power increases with increasing level of non-neutral genetic diversity (*Figure 2C*). However, predictive power is retained down to rather low pairwise distances, see *Figure 2—figure supplement 1*, where the model is a poor approximation. This suggests that the relationship between fitness and the structure of genealogical trees is more universal than the specific details of the mutation effect distribution that drive evolutionary dynamics (*Neher and Hallatschek, 2013*). The essence of this relationship between fitness and tree shape is picked up by the LBI. When applied to influenza A/H3N2 viruses sequences, a ranking by LBI predicts progenitor lineages with high accuracy.

One of the dominant paradigms for influenza A/H3N2 virus evolution has been the exploration of 'neutral' networks, punctuated by bursts of rapid adaptation through large effect mutations (*Koelle et al., 2006*; *Nimwegen et al., 1999*). In contrast, our ability to make meaningful predictions from the shape of genealogical trees of influenza virus sequences suggests that fitness variation persists in A/H3N2 populations. Fitness in the context of seasonal influenza viruses includes antigenic evolution as well as compensatory and deleterious mutations–within HA and other segments–that may contribute to fitness variation, shape the genealogies, and be determinants of future success. This conclusion is consistent with other existing evidence for ubiquitous selection in A/H3N2 populations (*Bhatt et al., 2011*; *Strelkowa and Lässig, 2012*). The applicability of our fitness inference scheme and the LBI ranking is further supported by the substantial enrichment in the number of non-synonymous substitutions at epitope loci in the lineages with predicted high relative fitness. These epitopes historically have high $dn/ds$ suggesting positive selection. Our model is agnostic to sequence and protein structure but nevertheless associates branches containing these mutations with increasing fitness.

It is also clear that large effect mutations, such as the ones associated with antigenic cluster transitions (*Koel et al., 2013*) can play an important role in the evolution of human seasonal influenza viruses. Many of the years in which our predictions are suboptimal (e.g., 1995, 2002, and 2004) correspond to

**Table 1.** Non-synonymous mutations at epitopes correlate with increasing fitness

| Quartile | # non-syn | # syn | # epi | # Koel |
|---|---|---|---|---|
| 25 | 130 | 155 | 43 | 7 |
| 50 | 159 | 178 | 57 | 10 |
| 75 | 184 | 205 | 74 | 21 |
| 100 | 209 | 222 | 115 | 22 |
| total | 682 | 760 | 289 | 60 |

| Comparison | enrichment | p-value |
|---|---|---|
| non-syn vs syn | 1.12 | n.s. |
| epi vs syn | 1.9 | 0.002 |
| Koel vs syn | 2.2 | 0.08 |
| epi vs non-syn | 1.7 | 0.015 |
| Koel vs non-syn | 2.0 | n.s. |

For each tree constructed for the years 1995–2013, we calculated the increment in $\lambda_i(\tau)$ with $\tau = 0.0625$ along each branch and determined the likely mutations on each branch. Branches were then sorted into quartiles according to changes in $\lambda_i(\tau)$. The left table shows the counts of non-synonymous (non-syn), synonymous (syn), non-synonymous mutations at epitope site (epi) and non-synonymous mutations at Koel positions (Koel) for branches in different quartiles. The right table quantifies the enrichment of certain types of mutations on branches in the top quartile relative the bottom quartile. Non-synonymous mutations at epitopes and Koel positions are approximately twofold enriched relative to synonymous mutations. Enrichment (odds ratio) and p-values were obtained using the Fisher exact test as implemented in scipy.stats (**Oliphant, 2007**).

antigenic cluster transitions in which antigenic properties changed drastically via specific large effect mutations. We tried to improve predictions by assigning additional positive fitness increments to substitutions at those loci identified by Koel et al. While this did improve results in some years, it also resulted in false positives which erased the overall improvement in predictive power. In some years in which these mutations are important, they tend to occur on many genetic backgrounds. This could explain why these mutations be themselves are not very predictive in our framework.

The fact that the branching patterns of reconstructed influenza A/H3N2 trees are predictive is surprising. In addition to occasional large effect effect mutations, e.g. those that cause substantial antigenic change, confounders such as the heterogeneity of sampling, complicated migration patterns, and demographic substructure should hamper prediction. The insensitivity to local oversampling is expected from the structure of our algorithm which senses the total length of sub-trees (rather then the number of leaves). Local oversampling will add many very short branches that perturb the total tree length only slightly. Subpopulations of different size, seasonality, and migration patterns, however, will perturb the coalescence patterns in parts of the reconstructed tree and should decrease predictability. Successful prediction therefore reinforces the conclusion that circulating influenza A/H3N2 populations harbor fitness variation. On the other hand, predictions might be improved by combining the shape of genealogical trees with antigenic information (**Bedford et al., 2014**), biophysical and structural knowledge (**Koel et al., 2013**), patterns of past evolution (**Łuksza and Lässig, 2014**), and plausible geographic sources (**Russell et al., 2008**; **Lemey et al., 2014**). However, each of these refinements introduces additional parameters into the model that need to be trained if not known a priori.

A defining feature of our method to predict evolution is that it can operate on a static set of sequences from a single time point and does not require historical data. We use historical data for influenza A/H3N2 only to validate the predictions. In **Figure 5**, we compare our results to a method that explicitly uses historical data (available for the influenza A/H3N2) to identify low frequency but expanding clades. By extrapolating their expansion into the future, one can anticipate the dominant strains of next year. Interestingly we found that prediction based on the reconstructed genealogy not only captures similar information, but also performs comparably if not better, even without access to historical data.

In summary, we have shown that the shape of reconstructed genealogies holds information about the relative fitness of the sampled individuals that can be exploited to predict the genetic composition of future populations, at least when fitness differences depend on multiple mutations. Since our algorithm requires nothing but a reconstructed genealogy as input, it should be applicable in many scenarios ranging from RNA viruses to cancer cell populations.

## Materials and methods

### Derivation of the fitness inference algorithm

Our algorithm is based on a branching process approximation to replicating clones within a finite population. Here, we first show how we use this approximation to calculate the probability that offspring

of an individual with a certain fitness are sampled. From there, we derive an equation for the branch propagators, that we solve numerically, and combine the propagators into the expression for the posterior fitness distribution given in *Equation (1)*.

## Offspring number distributions

The quantitative probabilistic description of clonal propagation is provided by the distribution $P(n|x,t)$ of the number of offspring $n$ after time $t$ given the ancestor had fitness $x$. Using a '1st-step' equation, that is, writing an equation for infinitesimal changes at the initial point $(y,t)$, we find for the backwards master equation for $P(n|x,t)$

$$P(n|x+\Delta tv, t+\Delta t) = [1 - \Delta t(2+x+u)]\, P(n|x,t) + \Delta t\langle uP(n|x+s,t)\rangle$$
$$+ \Delta t(1+x) \sum_{n'=0}^{n} P(n-n'|x,t)P(n'|x,t) \tag{2}$$

where the death rate is set to one and the birth rate is given by $1+x$ (see also (*Neher and Hallatschek, 2013*)). The first term corresponds to the probability of nothing happening in the time interval $\Delta t$ and the second term in $\langle \cdot \rangle$ corresponds to mutations averaged over the distribution $\mu(s)$ of possible fitness effects $s$ with the total mutation rate given by $u = \int ds\, \mu(s)$. The last term corresponds to replication of the individual. At the earlier time point $t+\Delta t$, fitness $x$ was larger by $\Delta tv$ due to the deterioration of the environment with velocity $v$. So far, this equation holds for arbitrary distribution of fitness effects. To make analytical progress, we assume that the distribution of mutational effects is short-tailed (exponential or steeper) and that the total mutation rate $u$ is large compared to the typical effect. In this case, *Equation (2)* can be rearranged into a differential equation where mutations are captured by the mean mutational effect and the mutational variance (*Tsimring et al., 1996*; *Cohen et al., 2005*; *Neher and Hallatschek, 2013*).

$$v\frac{\partial P(n|x,t)}{\partial x} + \frac{\partial P(n|x,t)}{\partial t} = -(2+x)P(n|x,t) + u\langle s\rangle\frac{\partial P(n|x,t)}{\partial x} + \frac{u\langle s^2\rangle}{2}\frac{\partial^2 P(n|x,t)}{\partial x^2}$$
$$+ (1+x)\sum_{n'=0}^{n} P(n-n'|x,t)P(n'|x,t) \tag{3}$$

The second term on the right hand side corresponds to the directional effect of mutations on fitness, while the third term to the diffusive dynamics of fitness due to mutations. To further analyze the behavior of $P(n|x,t)$, it is useful to consider the generating function $\psi_\omega(x,t) = \sum_n (1-\omega)^n P(n|x,t)$, which obeys

$$\frac{\partial \psi_\omega(x,t)}{\partial t} = -(2+x)\psi_\omega(x,t) + (u\langle s\rangle - v)\frac{\partial \psi_\omega(x,t)}{\partial x} + \frac{u\langle s^2\rangle}{2}\frac{\partial^2 \psi_\omega(x,t)}{\partial x^2} + (1+x)\psi_\omega^2(x,t) \tag{4}$$

Defining $\phi_\omega(x,t) = 1 - \psi_\omega(x,t)$, the fitness diffusion constant $D = u\langle s^2\rangle/2$, and the variance in fitness $\sigma^2 = v - u\langle s\rangle$, we have

$$\frac{\partial \phi_\omega(x,t)}{\partial t} = x\phi_\omega(x,t) - \sigma^2\frac{\partial \phi_\omega(x,t)}{\partial x} + D\frac{\partial^2 \phi_\omega(x,t)}{\partial x^2} - (1+x)\phi_\omega^2(x,t) \tag{5}$$

with initial condition $\phi_\omega(x,0) = \omega$. This equation for the generating function can be solved numerically or analytically in limiting cases. To approximate the fitness distribution on a given tree, we will solve this equation numerically.

It is also useful to explicitly define the 'reproductive value' $R(x,t)$ defined as the expected number of offspring of a genotype with fitness $x$ after $t$ generations, $R(x,t) = \sum_n nP(n|x,t)$. From the definition of the generating function it follows that $R(x,t) = \partial_\omega \phi_\omega(x,t)\big|_{\omega=0}$. Differentiating *Equation (5)* w.r.t. $\omega$ and noting that $\phi_\omega(x,t)\big|_{\omega=0} = 0$ yields a linear equation for $R(x,t)$ (essentially *Equation (5)* without the term $\phi^2$) which can be readily integrated. The expected number of offspring of one individual after time $t$ given it initially had fitness $x$ is

$$R(x,t) = e^{xt - \frac{\sigma^2 t^2}{2} + \frac{Dt^3}{3}} \qquad (6)$$

This approximation is only valid for times short compared to the coalescence time $T_c$, but it offers important insight into the dynamics of lineages: Initially, the lineage grows into a clone with rate $x$. The second term in the exponent describes how this growth slows since the remainder of the population is adapting with rate $\sigma^2$. The last term accounts for the fact that the offspring we consider can themselves change in fitness through mutations, the action of which is captured by the fitness diffusion constant $D$.

## Lineage sampling probability

The generating function $\phi_\omega(x,t)$ derived above has the interpretation of the probability that a lineage is represented in a sample of size $M$ from a population of size $N$ with $\omega = M/N$. From its definition, we have

$$\phi_\omega(x,t) = 1 - \sum_{n=0}^{\infty} P(n|x,t)(1-\omega)^n. \qquad (7)$$

Each term $(1-\omega)^n$ is the probability that none of the $n$ offspring are in the sample. By summing over the distribution of $n$ and subtracting the sum from 1, one obtains the probability of at least one offspring being sampled. The generating function can be accurately approximated in regimes where $\phi_\omega$ is small and the non-linear term in *Equation (5)* can be neglected, as well as the regime of large enough $x$ where $\phi$ 'saturates': $\phi_\omega(x,t) \approx x$, see (*Neher and Hallatschek, 2013*). These two asymptotic solutions can be combined to yield the approximation

$$\phi_\omega(x,t) \approx \frac{\omega x R(x,t)}{x + \omega[R(x,t) - 1]} \qquad (8)$$

Note that this approximation satisfies the initial condition $\phi_\omega(x,0) = \omega$, correctly tends to $x$ for $x > 0$ at long times, and recovers the neutral behavior $\phi_\omega(0,t) = \omega/(1 + \omega t)$ in the $x = \sigma^2 = D = 0$ limit.

## Branch propagator

Having calculated the lineage sampling probability, we are now in a position to derive equations governing the behavior of the branch propagator, that is, the probability of there being an individual with fitness $x$ at time $t'$ (the child), given it descends from an ancestor with fitness $y$ at time $t$ and all sampled descendants of the ancestor are also descendants of the child. The latter condition amounts to the requirement that in a tree the link between the ancestor and the child does not branch. Using a '1st-step' equation similar to *Equation (2)*, we have

$$g(x,t'|y + \sigma^2 \Delta t, t + \Delta t) = g(x,t'|y,t) - \Delta t(2+y)g(x,t'|y,t)$$
$$+ \Delta t D \frac{\partial^2 g(x,t'|y,t)}{\partial y^2}$$
$$+ \Delta t 2(1+y)[1 - \phi_\omega(y,t)]g(x,t'|y,t). \qquad (9)$$

The last term describes a 'birth' event in the ancestral lineage with one of the branches surviving up to $t'$ (at which time its fitness is in the $[x, x+dx]$ interval) while the other one is not sampled, which occurs with probability $1 - \phi_\omega(y,t)$ at a sampling density $\omega$. The $y \to y + \sigma^2 \Delta t$ shift in the argument of the term on the left-hand-side parametrizes the translation of the mean fitness in time $\Delta t$. *Equation (9)* reduces to the differential equation

$$\partial_t g(x,t'|y,t) = [y - 2\phi_\omega(y,t)]g(x,t'|y,t) - \sigma^2 \partial_y g(x,t'|y,t) + D\partial_y^2 g(x,t'|y,t) \qquad (10)$$

which is complemented with the initial condition $g(x,t|y,t) = \delta(x-y)$. In deriving this condition, we have assumed that $y \ll 1$, which is a good assumption when $\sigma$ (the standard deviation in fitness) is small. The fitness differences in a single generation are small in most populations, such that this assumption is not restrictive. Furthermore, violation of this assumption does not change the qualitative

behavior of the $g(\cdot|\cdot)$. When inferring fitness on trees, we will generally solve this equation numerically. Some limits, however, can be addressed analytically as we will see below.

Numerical solutions of $g(x,t'|y,t)$ are shown in **Figure 6**. For a fixed ancestor at $(y,t)$, $g(x,t'|y,t)$ is the density of offspring with fitness $x$ at time $t'$ subject to the following condition: Only one individual from this group of offspring contributes to the sample at present (this is the condition that the lineage connecting $(x,t')$ and $(y,t)$ is unbranched). The propagator $g(x,t'|y,t)$ broadens in $x$ as $t-t'$ increases as shown in **Figure 6A** for a case of high (red, $y>2$) and low (blue, $y=0$) initial fitness. **Figure 6B** shows how the integral $\int dx\, g(x,t'|y,t)$ increases with $t$ for $y>0$ but decreases for $y<0$. The integral of $\int dx\, g(x,t'|y,t)$ differs from the reproductive value $R(y,t-t')$, shown as dashed lines in **Figure 6B**, only in the additional sampling condition.

At fixed $(x,t')$, $g(x,t'|y,t)$ is peaked around $x$ for small $t-t'$ and this peak move to higher fitness as as $t-t'$ increases and converges against a steady distribution far in the past. This is seen in **Figure 6C**, where the $g(x,t'|y,t)$ is plotted as a function of $y$. Far in the past $g(x,t'|y,t)$ has a well defined maximum at $y\approx 3\sigma$. This steady distribution is shaped by two opposing trends: Fit ancestors (large $y$) leave more offspring and are hence more likely sampled. Too fit ancestors, on the other hand, should leave many individuals at time $t'$ that ultimately contribute to the sample. The width of the steady state distribution is determined the diffusion constant $D$.

As a special case, we will sometimes be interested in a *terminal* branch propagator, which takes the lineage all the way to the present generation, $t'=0$. Marginalizing and multiplying by the sampling probability $\omega=M/N\ll 1$ defines the probability of the $(y,t)$ ancestor to be a direct progenitor of a sampled genome: $G(y,t)=\omega\int dx\, g(x,0|y,t)$. Interestingly, for positive $y$, one expects this probability to initially increase with increasing $t$ because the reproductive value - i.e. expected number of surviving offspring - for relatively fit individuals increases with time, so that their offspring constitute a larger fraction of the population and are therefore more likely to appear in the sample. At longer times however $G(y,t)$ is expected to start decreasing, because it is increasingly unlikely that the lineage emanating from a highly fit ancestor far in the past, remains unbranched (i.e., has only a single descendant in the sample).

For small times and moderate parental fitness $y$, the term enforcing non-branching in **Equation (10)** can be neglected. In this case, the terminal branch propagator simplifies to

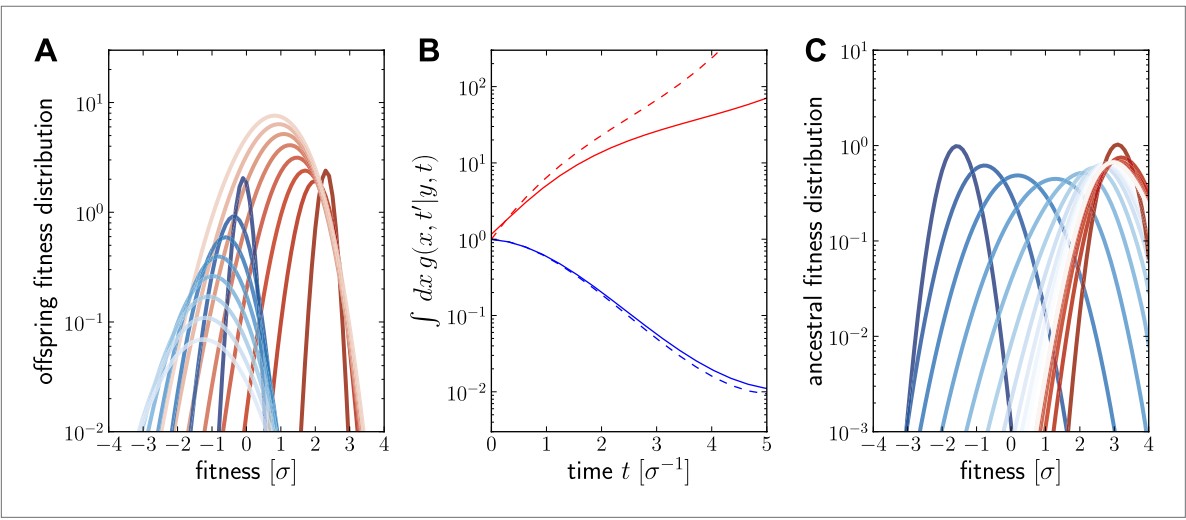

**Figure 6**. Numerical solution for the lineage propagator. Panel **A** shows $g(x,t'|y,t)$ as a function of $x$ for different $t'$ at $t=0$ given the ancestor had Malthusian fitness $y=0$ (blue) or approximately $y=2\sigma$ (red). In both cases, the offspring tend to get less fit and the distribution broadens due to additional mutations. Saturated colors correspond to small $t-t'$, light colors large $t-t'$. Panel **B** shows $\int dx\, g(x,t'|y,t)$ as a function of $t-t'$ for the high (red) and low (blue) fitness ancestor. The dashed lines show the approximation given in **Equation (6)**. In the high fitness case, **Equation (6)** overestimates $\int dx\, g(x,t'|y,t)$ since it does not account for the non-sampling contribution. Panel **C** shows $g(x,t'|y,t)$ as a function of $y$, given the offspring is unfit (blue) or fit (red). Ancestors tend to be fit regardless of offspring fitness and both ancestral distributions converge to a common curve far back in time.

$$G(y,t) \approx e^{yt - \frac{\sigma^2 t^2}{2} + \frac{Dt^3}{3}} \tag{11}$$

and is hence identical to the reproductive value *Equation (6)*.

## Tree-based inference

Armed with branch propagators we can now write down a joint probability of ancestral fitness on any given tree. Let $x_i$ denote the fitness of node $i$ starting with $i = 0$ at the root of the tree, $i = 1, \ldots, n_{int}$ for internal nodes, and $i = n_{int} + 1, \ldots, n_{int} + n_{ext}$ for external nodes. Furthermore, denote the children of node $i$ by $i_j$, where $j$ runs over the number of children. The joint probability distribution of all nodes in the tree is then given by

$$P(\mathbf{x}|T) = \frac{p_0(x_0)}{Z(T)} \prod_{i=0}^{n_{int}} \prod_j g\left(x_{i_j}, t_{i_j} \middle| x_i, t_i\right) \tag{12}$$

where $Z(T)$ is a normalization factor, $p_0(x)$ is the fitness distribution in the population, and the second product runs over all $j$ children of node $i$. In contrast to *Equation (1)*, *Equation (12)* allows for polytomies in the tree. In writing down *Equation (12)*, we have made the approximation that the total population size is unconstrained and that different branches of the tree do not interact. In populations dominated by selection, this is a good approximation since coalescent properties depend only weakly on the population size.

This joint probability lives in a too high dimensional space to be practically useful, however, the tree structure makes it easy to marginalize the distribution. We commence 'integrating out' the independent fitness variables of the leaves, followed by integrating over the fitness values of the parents of these leaves until we arrive at the root of the tree. This defines an iterative 'message passing' process (*Mézard and Montanari, 2009*) in which the 'message' node $i$ sends to its parent $p_i$ is calculated via

$$m_{\uparrow i}(x_{p_i}) = \int dx_i \; g\left(x_i, t_i \middle| x_{p_i}, t_{p_i}\right) \prod_j m_{\uparrow i_j}(x_i) \tag{13}$$

where the product is over all children $j$ of node $i$ (note that the times $t_i$ and $t_{p_i}$ are fixed properties of the tree). For terminal nodes $i$ without children, $m_{\uparrow i}(x_{p_i})$ is simply the terminal branch propagator. Similarly, we calculate "messages" passed downstream to child $j$ of node $i$:

$$m_{\downarrow i_j}(x_{i_j}) = \int dx_i \; g\left(x_{i_j}, t_{i_j} \middle| x_i, t_i\right) m_{\downarrow i}(x_i) \prod_{k \neq j} m_{\uparrow i_k}(x_i) \tag{14}$$

The integrand is the product of the downstream message from the parental node and the upstream messages from all children of node $i$ other than child $j$. This product is further multiplied by the branch propagator to child $j$ and integrated over the fitness of node $i$.

Having calculated the up and down messages for each branch, we can simply calculate the marginal distributions of fitness $x_i$ by multiplying all messages going into a node $i$.

$$p(x_i) = \frac{1}{Z_i} m_{\downarrow i}(x_i) \prod_j m_{\uparrow i_j}(x_i) \tag{15}$$

where $Z_i$ assures normalization. Our inference uses the mean marginal fitness to rank internal and external nodes.

For a pre-terminal node, the 'up-message' (*Equation (13)*) involves multiplying the terminal branch propagators of all its children. If the node is recent, we can use approximation *Equation (11)* and obtain

$$m_{\uparrow i}(x_{p_i}) \sim \int dx_i \; g\left(x_i, t_i \middle| x_{p_i}, t_{p_i}\right) e^{T_{tot} x_i}, \tag{16}$$

where $T_{tot}$ is total tree length downstream of node $i$, which polarizes the fitness of node $i$ towards the high fitness edge. For a given number of descendants, this total tree length is maximized by a star topology. This corresponds to recent findings that multiple mergers in genealogies are associated with

rapid expansion of clones founded by exceptionally fit individuals (**Brunet et al., 2007**; **Desai et al., 2013**; **Neher and Hallatschek, 2013**).

## Calculating the local branching index (LBI)

The LBI defined as the integrated exponentially discounted tree length surrounding a node can be calculated in a very similar way to the message passing framework used above to evaluate the fitness distributions. The corresponding 'up'-messages to the parent of node $i$ is simply

$$m_{\uparrow i} = \tau \left( 1 - e^{-b_i/\tau} \right) + e^{-b_i/\tau} \sum_j m_{\uparrow i_j} \tag{17}$$

where $b_i$ is the branch length of node $i$ and the sum runs over the children $i_j$ of node $i$. Similarly, the down message from a parent $i$ to child $i_j$

$$m_{\downarrow i_j} = \tau \left( 1 - e^{-b_{i_j}/\tau} \right) + e^{-b_{i_j}/\tau} \left[ m_{\downarrow i} + \sum_{k \neq j} m_{\uparrow i_k} \right] \tag{18}$$

After having calculated all up and down messages, the exponentially discounted tree length is given by

$$\lambda_i(\tau) = m_{\downarrow i} + \sum_j m_{\uparrow i_j} \tag{19}$$

## Implementation of the inference algorithm

The fitness inference algorithm is implemented in Python using the libraries SciPy and NumPy (**Oliphant, 2007**). Roughly, we have implemented one class, survival_gen_func, that integrates the fitness propagator on a discrete fitness grid. This class is used by the class fitness_inference to calculate the marginal distribution of fitness at each external and internal node of a given tree. The calculation of the marginals is done using a message passing approach (**Mézard and Montanari, 2009**). This fitness inference class is then subclassed to accommodate influenza specific features. All code associated with this manuscript is available at https://github.org/rneher/FitnessInference.

To predict the sequence closest to the future population in a multiple sequence alignment, we build a maximum likelihood tree using fasttree (**Price et al., 2009**) (the fasttree code was modified slightly to resolve short branches better). The reconstructed tree was passed to the fitness inference class. Following fitness inference, internal or external nodes were ranked by their expected fitness and we report the top ranked node as our prediction.

The branch propagator depends on fitness diffusion constant $D$, the standard deviation in fitness $\sigma$, and the sampling fraction $\omega$. For the numerical implementation, we measure time in unites of $\sigma^{-1}$ and selection strength in units of $\sigma$ and the dimensional fitness diffusion constant is $\Gamma = D\sigma^{-3}$. The initial condition for the generating function is $\phi_\omega(x, 0) = \omega/\sigma$ in these units.

In order to apply our algorithm to a tree reconstructed from sequences, we need to convert branch length into time in units of $\sigma^{-1}$. Given an alignment, we can calculate the average pairwise nucleotide distance $\pi \approx 2\mu\langle T_2 \rangle$, where $\langle T_2 \rangle$ is the average pair coalescent time and $\mu$ is the per site mutation rate. For an adapting population in the SBD model, we have $\langle T_2 \rangle \sigma \approx \Gamma^{-1}$ (**Neher and Hallatschek, 2013**). Given a choice for $\Gamma$, the conversion factor $\beta$ from nucleotide distance to $\sigma^{-1}$ units is determined by

$$\frac{\pi}{2\beta} = \frac{1}{\Gamma} \quad \Rightarrow \quad \beta = \frac{\Gamma\pi}{2}. \tag{20}$$

In addition to estimating fitness from the tree, we also measure the frequency changes of clades over time. For influenza A/H3N2 virus data, we partition sequences into three intervals of equal length between May and February and calculate the fraction of sequences that are below every internal nodes in each of these intervals (using a pseudocount of 5). From these three frequency values, we estimate the expansion rate by fitting a line to the logarithm of the frequencies.

## Simulations

We use the population genetics library FFPopSim (**Zanini and Neher, 2012**) to implement an individual based simulation with fixed fitness variance $\sigma = 0.03$. Mutations are introduced at random sites

in random individuals with rate $\mu$. We varied the total genomic mutation rate $u = L\mu$ between 0.016 and 0.256, where the total number of simulated sites is $L = 2000$. Mutations at all sites are by default deleterious, with effects drawn from an exponential distribution. To emulate a changing environment, we redraw the fitness effect of random positions within the first 500 sites at random with a total rate of $n_A = 0.02, \ldots, 0.16$ per generation. Beneficial effects are drawn from a gamma distribution with shape parameter 2 and the same scale as the deleterious mutations. Every 200 generations, a random sample of 200 sequences is written to file and later used to predict the sequence closest to the next sample. The simulation code is provided as flusim.cpp in the above mentioned repository.

### Influenza data

All sequences of influenza A/H3N2 viruses from human hosts from 1968 to 2014 that cover the entire HA1 domain were downloaded from IRD and aligned using the alignment feature provided by IRD with default settings (*Squires et al., 2012*). The alignment was inspected by eye and trimmed to the HA1 domain. A few obvious outliers, lab strains, and sequences with indels or more than 4 ambiguous nucleotides were removed manually. For each strain the location information was converted to longitude and latitude at the country level and the strain was classified into rough geographic regions based on longitude and latitude. Only sequences with geographic information at the country level and date information with at least month accuracy were used. To avoid sampling bias, we subsampled the data to at most 100 sequences from either North America and Asia and used repeated subsamples to assess the robustness of the predictions. In years where less than 100 sequences are available from one of the geographic regions, we repeatedly used 70% of the available data. Increasing the sample size has negligible effect on prediction accuracy beyond a sample size of 100.

## Acknowledgements

We are grateful to Michael Elowitz, Paul Rainey and Eric Siggia for critical reading of the manuscript.

## Additional information

### Competing interests

RAN: Reviewing editor, *eLife*. The other authors declare that no competing interests exist.

### Funding

| Funder | Grant reference number | Author |
| --- | --- | --- |
| European Research Council | ERC-Stg-260686 | Richard A Neher |
| Royal Society | University Research Fellowship | Colin A Russell |
| National Institutes of Health | R01 GM086793 | Boris I Shraiman |

The funders had no role in study design, data collection and interpretation, or the decision to submit the work for publication.

### Author contributions

RAN, Conception and design, Acquisition of data, Analysis and interpretation of data, Drafting or revising the article; CAR, BIS, Conception and design, Analysis and interpretation of data, Drafting or revising the article

### Author ORCIDs

Richard A Neher, http://orcid.org/0000-0003-2525-1407

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
