## [Decision Letter]

Thank you for sending your work entitled “Predicting evolution from the shape of genealogical trees” for consideration at *eLife*. Your article has been favorably evaluated by Chris Ponting (Senior editor) and 3 reviewers, one of whom is a member of our Board of Reviewing Editors.

The Reviewing editor and the other reviewers discussed their comments before we reached this decision, and the Reviewing editor has assembled the following comments to help you prepare a revised submission.

All reviewers agreed that inferring fitness from a phylogeny is an interesting and promising approach. The principal innovations were considered to be: (a) The likelihood function, which is an approximation to a birth-death process with variable (and evolving) birth rate; (b) the evaluation of the method through simulation; (c) the application to the influenza data sets, showing the information about fitness inherent in tree shape; (d) the comparison of the method to the recent work from Luksza and Lassig, which focused on key sites in key immunological proteins; and, (e) the elaboration of the method to include addition information about sites of known importance and longitudinal information within a year that can help spot growing clades.

Nevertheless, there was no clear consensus among the reviewers with regard to its suitability for publication and they would like to consider a revised version of the manuscript, which should not require much extra work but should adequately address the three comments and criticisms below.

1) The conclusion of many minor-effect mutations will need to be supported by better evidence or to be formulated differently. Whilst you claim that the results support the notion that a substantial fraction of adaptive evolution is through a quantitative (infinitessimal-style) response, it appears feasible that the primary driver of genealogical history/selection is indeed through HA/NM, but acts in a structured population. If so, and there is a semi-neutral accumulation of mutations along the branches, the genealogy becomes a proxy for the relative fitness of cryptic sub-populations within the species. The fact that the Luksza and Lassig results are so similar suggests that the two approaches take advantage of very similar information. The reviewers were also sceptical about the inclusion of Koel mutations into the prediction scheme. The sites of these mutations have been identified in a very recent publication through their importance for antigenic substitutions (15). Hence, for most of the prediction period, they introduce posterior information into the prediction; this caveat should at least be noted. We ask you to consider including results for the model with temporal information (clade growth rates), but without the Koel term; this will quantify the contribution of that term to the full prediction. If the net contribution of the Koel term remains limited, the message of the paper might become stronger by making exactly that point. This is of relevance for influenza research because the current analysis focuses to a large extent on the identification of few large-effect antigenic changes. Finally, please comment further on the inference of small effects: it may be circular since it is an input assumption of your method.

2) Inadequate discussion of model assumptions and impact of data structure (temporal spread, spatial structure, sampling inhomogeneities). (a) The model contains a parameter lambda that scales the branch lengths of the coalescent tree. Is this a canonical scale parameter within the traveling wave theory or a heuristic extension? If so, it should be marked as such. (b) Similarly, is the form of the propagator for branches with Koel mutations, [Disp-formula equ2], supported by the theory with heterogeneous effects or a heuristic? (c) The additional weighing of clade frequency changes (growth rates) rho_i via [Disp-formula equ3] appears to us foreign to the coalescent approach, which should predict this change rather than using it as a separate input. In other words, we would expect the frequency change term to be either redundant or to measure deviations from the coalescent model. The relationship between these two model components should be explained and quantified by a scatter plot showing coalescent-predicted (from the two-parameter model) vs. measured clade growth rates rho_i. (d) We noted that the best model for influenza, which includes the Koel reward and the clade growth term, uses four parameters; this should be made explicit in the main text. (e) Can one relate the fitness diffusion constant D to standard population-genetic parameters, such as the rate and effect distribution of beneficial mutations? Note that the mean coalescent time T2 and the speed of the fitness wave (via the average lifetimes of polymorphisms destined for fixation?) can at least roughly be estimated from the data. Are these estimates consistent with the model input parameters?

Furthermore: (f) What are the effects of sampling inhomogeneities on the tree, as they exist in the influenza case? If a given clade is oversampled, this may produce a spurious fitness signal in the authors' method. To what extent is the method robust to such fluctuations, and where are the limits of applicability? (g) What are the effects of the temporal spread of input data within a given year? Note that the influenza data are strains from a full year; they deviate from the model assumption of input at a given point in time. How strongly? i.e., how does this time interval compare to the other characteristic time scales of the problem? The effects of these inhomogeneities on model predictions can best be assessed by simulations of the kind already performed in this work and described in the results Section. (h) It may be worth stressing that the prediction scheme works for a limited time into the future. This time can probably be estimated from the model parameters (fitness diffusion constant...). It would be interesting to quote this prediction time for influenza.

3) Whilst the motivation of the paper is intuitive and the paper easy to understand for non-experts, this simplicity masked many assumptions of the model which was considered problematic. For example, the Methods section should explain more fully the inference algorithm (Neher and Hallatschek), especially its critical assumptions. This is important because the predictions on simulated data can be rather bad or even misleading (see Figure 2 low mutation rates), and it will be important to assess its applicability to other data. In addition, in deriving [Disp-formula equ10], it seems that there is an assumption of small fitness effects (x approx 0), such that (1-x) phi^2 = phi^2. On the other hand, you state that for large enough x, phi = x. Then you plot fitnesses in the range (-4 sigma, 4 sigma) (Figure 4). Does this put a very stringent restriction on sigma^2? It is possible that there is a mathematically solid justification for this, but you should make it easier for the reader to understand. Finally, in the methods (and as discussed above under point 2), you also briefly comment on how your assumption of “non-sampling” affects fitness estimates. What does this mean for the sample sizes in the case of influenza? Is it necessary to use a small sampling fraction to obtain a reliable prediction? Also, would sampling bias have a large effect on the predictions. If a certain clade is over-represented in the sample, the algorithm would infer higher fitness for that clade, correct?

---

## [Author Response]

The main points raised during the review were (i) the suggestion that small effect mutations are important for influenza A/H3N2 adaptation, and (ii) model assumptions, parameter dependence, and lack of intuition for the inference algorithm.

To address the first point, we examined the dependence of predictive power on the number of mutations contributing to fitness variation and included this as a supplement to Figure 2. While predictive power increases when more mutations with smaller effects contribute, predictability is retained down to relatively small number of mutations contributing to fitness differentials. Prediction capacity is retained because Selection-biased Diffusion, which describes fitness dynamics along lineages in the “infinitesimal” limit (of many small effect mutations), holds approximately even in the case when only a few mutations contribute.

Predictability alone therefore not an unequivocal argument for many small effect mutations. Nevertheless, predictability requires that the population has persistent fitness variation distributed over a number of loci, in effect behaving closer to the infinitesimal case, than to the case of periodic sweeps. We have adapted the manuscript to reflect this insight.

To address the concerns regarding model assumptions, parameter fitting, and the lack of intuitive insight into the inference algorithm, we now discuss in greater detail the analytic result that downstream tree length is the most important determinant of fitness of young nodes. We extended this argument for all nodes on the tree and introduced a local branching index (LBI) defined as the length of the tree surrounding a node in its neighborhood (implemented as exponential weighting). Local tree length increases with branching, which in turn is indicative of high fitness. This intuitive connection captures the essence of the fitness inference algorithm. Moreover, the LBI predicts the progenitor lineages almost as well as the full probabilistic fitness inference when applied to simulation data. We used simulation data to fix the ‘only’ free parameter of the LBI – the size of the neighbourhood – and applied it to influenza without any fitting parameters whatsoever. The predictions obtain this way are almost as good as those obtained previously with the full fitness inference. Since that latter required choosing at least 2 parameters, we feel that the LBI is superior in practice even though it misses one year where the fitness inference did well (1996: there is little data in this early year anyway).

To accommodate these improvements, we have included a section on the LBI and redid the influenza analysis using the LBI to rank isolates. All conclusions remain the same, but the nature of fitness inference and progenitor prediction have become much more transparent through the identification of the local tree length as the essence of fitness prediction. We have streamlined the manuscript and removed the discussion of the gamma parameter (no longer needed when using the simpler predictor) and no longer show the results including the Koel mutations (which didn't add much to the predictions anyway).

We included one additional year (2013) of influenza predictions, as data to evaluate this prediction have become available since our initial submission (both the full fitness inference and the LBI predict this year well). In addition, we have further streamlined and commented the code that has been deposited on github. We feel that these revisions have greatly improved our manuscript. We provide a point by point response to all referee comments below.

*1) The conclusion of many minor-effect mutations will need to be supported by better evidence or to be formulated differently. Whilst you claim that the results support the notion that a substantial fraction of adaptive evolution is through a quantitative (infinitessimal-style) response, it appears feasible that the primary driver of genealogical history/selection is indeed through HA/NM, but acts in a structured population. If so, and there is a semi-neutral accumulation of mutations along the branches, the genealogy becomes a proxy for the relative fitness of cryptic sub-populations within the species*.

In the previous version of the manuscript, we stated that predictability of influenza using a model assuming many small effect mutations suggests that influenza A/H3N2 evolution is in part dominated by such dynamics. We now quantify, using simulations, how predictability varies with the number of segregating mutations in the sample (Figure 2—figure supplement 1). Good predictions require several segregating mutations, but some predictability is retained down to low numbers. Hence, we cannot provide a reliable lower bound on the number of mutations contributing to fitness in any given year and we have reworded the relevant parts of the text to stress that our ability to make meaningful predictions stems from persistent variations in virus fitness.

If we understand the alternative scenario correctly, the structured population would merely reduce competition between different lineages on time scales shorter than the one year (H3N2 viruses do not persist locally between epidemics). Without persistent heritable differences between strains, we do not see how it would be possible for our method to make meaningful predictions. Furthermore, the 2-fold enrichment of nonsynonymous mutations at 50 epitope positions on branches with high inferred Delta fitness suggests that our algorithm picks up a genetic signature.

*The fact that the Luksza and Lassig results are so similar suggests that the two approaches take advantage of very similar information*.

Luksza and Laessig include an ad-hoc predictor meant to account for nonlinear or epistatic effects that counts the number of synonymous mutations below an internal node. This component of their model accounts for a substantial fraction of the predictive power. Incidentally, the number of synonymous mutations is a proxy for the total branch length below a node, which is intimately connected to the inferred fitness of internal nodes. The addition of the local branching index makes this essential feature of fitness prediction explicit.

*The reviewers were also sceptical about the inclusion of Koel mutations into the prediction scheme. The sites of these mutations have been identified in a very recent publication through their importance for antigenic substitutions (*[15]*). Hence, for most of the prediction period, they introduce posterior information into the prediction; this caveat should at least be noted. We ask you to consider including results for the model with temporal information (clade growth rates), but without the Koel term; this will quantify the contribution of that term to the full prediction. If the net contribution of the Koel term remains limited, the message of the paper might become stronger by making exactly that point. This is of relevance for influenza research because the current analysis focuses to a large extent on the identification of few large-effect antigenic changes*.

We agree that inclusion of the Koel mutation introduces after-the-fact information into our prediction. The predictive power of the Koel mutations is limited and improvement restricted to a few years. In some years, predictions become worse because Koel mutations appear multiple times on the tree and are mostly false leads. We did not claim Koel mutations to be predictive, but merely sought to show how flu specific knowledge can be combined with our general inference scheme. We now clearly state that within our framework the predictive power gained from looking at the occurrence of Koel mutations is not worth the extra parameter needed to include them (and have removed the predictions utilizing them). We still discuss the Koel mutations as potential large effect mutations beyond the scope of our model.

*Finally, please comment further on the inference of small effects: it may be circular since it is an input assumption of your method*.

As pointed out above, we now emphasize that persistent fitness variation is a prerequisite for prediction but that we cannot put a lower bound on the number of mutations contributing to fitness. We added a supplementary figure showing how predictive capacity depends on the genetic diversity in the sample (Figure 2—figure supplement 1).

*2) Inadequate discussion of model assumptions and impact of data structure (temporal spread, spatial structure, sampling inhomogeneities). (a) The model contains a parameter lambda that scales the branch lengths of the coalescent tree. Is this a canonical scale parameter within the traveling wave theory or a heuristic extension? If so, it should be marked as such*.

The parameter gamma has become obsolete since we now use the simpler algorithm (Local Branching Index) to rank influenza sequences. Within the selection-biased diffusion model, the conversion of branch length to time is fixed by theory and gamma=1. LBI has a single phenomenological parameter (setting the local scale) which we choose by comparing LBI to the full inference algorithm on simulated data.

*(b) Similarly, is the form of the propagator for branches with Koel mutations,*
[Disp-formula equ2]*, supported by the theory with heterogeneous effects or a heuristic*?

The parameter for branches with Koel mutation was ad-hoc. It could be made more principled, but this would involve an additional integration over the possible times at which this mutation could have arisen. Given that the value of the Koel mutations for prediction seems limited, we have removed this altogether.

*(c) The additional weighing of clade frequency changes (growth rates) rho_i via*
[Disp-formula equ3]
*appears to us foreign to the coalescent approach, which should predict this change rather than using it as a separate input. In other words, we would expect the frequency change term to be either redundant or to measure deviations from the coalescent model. The relationship between these two model components should be explained and quantified by a scatter plot showing coalescent-predicted (from the two-parameter model) vs. measured clade growth rates rho_i*.

The fitness inference or LBI can indeed be used to predict clade expansion into the next season. We included a figure supplement to Figure 4 (formerly Figure 3) that shows how highly ranked clades tend to expand.

We have also included a comparison of our predictions with predictions based solely on growth rate and another naive predictor based on ladderization of the tree. Neither of these predictors come close to the predictive power of our approach, likely because both of them are highly sensitive to sampling biases while our approach is much less so.

*(d) We noted that the best model for influenza, which includes the Koel reward and the clade growth term, uses four parameters; this should be made explicit in the main text*.

We no longer use this model as the improvement of the predictions did not warrant the two extra parameters. We explicitly discuss the trade-off between improved predictions and additional parameters. Our simplified local Branching Index ranking has only single parameter (the neighbourhood size), which we choose based on simulated data.

*(e) Can one relate the fitness diffusion constant D to standard population-genetic parameters, such as the rate and effect distribution of beneficial mutations? Note that the mean coalescent time T2 and the speed of the fitness wave (via the average lifetimes of polymorphisms destined for fixation?) can at least roughly be estimated from the data. Are these estimates consistent with the model input parameters*?

Yes, the fitness diffusion constant has a straightforward interpretation in terms of mutations rates and fitness effects. D is half the product of the mutation rate and the second moment of the effect size of mutations. Since time is measured in units of 1/\sigma, both the mutation rate the effect size are also measured in units of sigma. We now explicitly state this. In units of sigma, \Gamma = D\sigma^{-3} is proportional to the inverse sqrt(log N), which varies only slowly.

*Furthermore: (f) What are the effects of sampling inhomogeneities on the tree, as they exist in the influenza case? If a given clade is oversampled, this may produce a spurious fitness signal in the authors' method. To what extent is the method robust to such fluctuations, and where are the limits of applicability*?

Sampling can affect the fitness inference in exactly the direction suggested. But as we now discuss, our algorithm and the LBI are fairly insensitive to sampling biases. Since the algorithm senses the total length of subtrees, local over sampling and the addition of many similar sequences (with short branches) has little impact on our inferences. We tried to avoid sampling biases by using at most 100 sequences from Asia or North America – Asia being the primary source region for seasonal H3N2 viruses and North America being a sink region. More sophisticated corrections for sampling biases (i.e. by factoring in surveillance efforts in different countries) could be envisioned, but require additional data currently not available to us.

*(g) What are the effects of the temporal spread of input data within a given year? Note that the influenza data are strains from a full year; they deviate from the model assumption of input at a given point in time. How strongly? i.e., how does this time interval compare to the other characteristic time scales of the problem? The effects of these inhomogeneities on model predictions can best be assessed by simulations of the kind already performed in this work and described in the results Section*.

We had already tested the effect of continuous sampling in simulation data. The results are now included as Figure 2—figure supplement 2 and are very similar to data sampled from only one time point. Our influenza sequences come from a 10 month interval from May to February.

*(h) It may be worth stressing that the prediction scheme works for a limited time into the future. This time can probably be estimated from the model parameters (fitness diffusion constant ...). It would be interesting to quote this prediction time for influenza*.

We now clearly state that the scope of our method is predicting a progenitor of the future, rather than predicting future evolution. The progenitor lineage is independent of the time horizon on which the prediction is evaluated. On very short time scales (a few months), predicting clade growth rather than progenitors might be more appropriate.

*3) Whilst the motivation of the paper is intuitive and the paper easy to understand for non-experts, this simplicity masked many assumptions of the model which was considered problematic. For example, the Methods section should explain more fully the inference algorithm (Neher and Hallatschek), especially its critical assumptions. This is important because the predictions on simulated data can be rather bad or even misleading (see*
Figure 2
*low mutation rates), and it will be important to assess its applicability to other data*.

We have extended the Methods section and included relevant details and assumptions. The lower quality of predictions at low mutation rates stems from the fact that fitness diversity in the population depends on few mutations. The resulting granularity of the fitness distribution (and the large effect size of mutations) make prediction difficult. We have extended the discussion of the toy data results, included a figure supplement that explicitly shows the quality of fitness inference as a function of the genetic diversity in the sample. Furthermore, we point out the crucial assumptions when deriving the fitness inference algorithm.

*In addition, in deriving*
[Disp-formula equ10]*, it seems that there is an assumption of small fitness effects (x approx 0), such that (1-x) phi^2 = phi^2. On the other hand, you state that for large enough x, phi = x. Then you plot fitnesses in the range (-4 sigma, 4 sigma) (*Figure 4*). Does this put a very stringent restriction on sigma^2? It is possible that there is a mathematically solid justification for this, but you should make it easier for the reader to understand*.

We do indeed make an assumption of small x, that fitness differences in one generation are assumed to be <<1. Influenza lineage turnover happens on the scale of 1 to 2 years, suggesting that fitness differential in one generation (a few days) is on the order of a few percent. Even if this assumption is not justified, the qualitative behavior of all equations remains unchanged. The high fitness tail of phi changed from phi∼x to phi∼x/(1+x). We now point the constraints on the strength of selection, i.e., sigma, more carefully.

*Finally, in the methods (and as discussed above under point 2.), you also briefly comment on how your assumption of “non-sampling” affects fitness estimates. What does this mean for the sample sizes in the case of influenza? Is it necessary to use a small sampling fraction to obtain a reliable prediction? Also, would sampling bias have a large effect on the predictions. If a certain clade is over-represented in the sample, the algorithm would infer higher fitness for that clade*, *correct?*

The results depend only very weakly (square root of a logarithm) on the sampling size entering the non-sampling factor. Increasing omega (the assumed sampling fraction) by a large factor pushes all fitness estimates down, but has little effect on the relative ranking. When chosen correctly, the fitness estimates of terminal nodes should scatter around 0 consistent with the population distribution. For the simulated data, we know how to set omega, for the influenza data we now use the LBI to rank sequence which does not depend on the sampling fraction. As discussed above, our algorithm is fairly insensitive to sampling bias and we try to avoid sampling bias by down sampling that data to similar numbers of sequences from different geographic regions.